# FIT-LoRA: FIT YOUR LoRAs TO PRUNED LLMs WITHOUT ADDITIONAL TRAINING OR DATA

## ABSTRACT

Personalization of LLMs via fine-tuning has become a popular way to enhance performance on downstream tasks. However, the model adaptation obtained after fine-tuning is specific to the base model. Any modifications made to the structure of the base model require users to fine-tune on the downstream task again. During deployment, a base model may be modified using pruning to obtain several LLM scales tailored to specific compute requirements. In this scenario, it becomes challenging to keep up with personalization, since each derived model must be individually fine-tuned. To address this challenge, we explore the possibility of leveraging the base model's fine-tuned knowledge to personalize any derived models. In this paper, we present FIT-LoRA, a framework that enables fine-tuning knowledge transfer between a base LLM and derived LLMs of smaller scales without needing any training or access to the original fine-tuning data. We validate our approach by conducting extensive experiments covering representative datasets such as *BoolQ*, *SST-2*, *MRPC*, *RTE*, and *WinoGrande* across various model architectures, including Llama-2, Llama-3.1, Mistral, and Gemma-2. Furthermore, we show the effectiveness of our approach by demonstrating its compatibility across multiple types of state-of-the-art LLM pruning methods, including depth pruning, structured pruning, and sparsification.

## 1 INTRODUCTION

The remarkable progress in the development of Large Language Models (LLMs) has led to great achievements in general language tasks across diverse domains and tasks. LLMs can further benefit from fine-tuning on domain-specific tasks, where specialized or context-specific knowledge is essential. While pretraining provides a strong baseline, fine-tuning is important for many practical use cases. However, modern LLMs have scaled beyond billions of parameters, which has substantially increased the computational and memory requirements for training and serving such models.

To make LLMs more accessible and reduce the computational burden of training models, there has been efforts to work on parameter-efficient fine-tuning techniques (PEFT) (Han et al., 2024). Among them, Low-Rank Adaptation (LoRA) (Hu et al., 2022) is a widely used method. In parallel, there has been a significant research effort in the area of LLM compression. Compressed models enable LLMs to be embedded in more devices and reach more users during deployment. However, as the number of models needed for flexible deployment increases, so does the amount of fine-tuning that is required. This is because the fine-tuning process is tightly coupled to the base model. This makes it difficult to keep up with personalization as models undergo modifications.

Beyond the computational and logistical burden, updates made to a base model also present another challenge: the potential lack of access to the original dataset that was used for fine-tuning. In fields such as healthcare, finance, or legal, it may be impractical for the user to have repeated access to the dataset required for personalization. In an online system, training data may also arrive in a stream, and is not persisted for later use. Additionally, downstream users of the LoRA adapter (e.g., adapters from platforms like HuggingFace) may want to use the adapter on a compressed version of the base model. However, they do not have access to the training data or may not have resources for fine-tuning. These realities necessitate approaches that can capture and transfer learned knowledge without relying on repeated exposure to the original data.

We outline possible workflows as follows. For example, as illustrated in Figure 1(a), a base model is downstream-fine-tuned on proprietary data by a cloud provider, resulting in a LoRA adapter (highlighted in green). If an end user wishes to reuse this adapter on a compressed version of the base model (in light blue), the LoRA adapter is incompatible and must be fine-tuned again (in yellow). A possible choice could be to merge the adapter first and then compress, as illustrated by Figure 1(b). However, this would create a separate model per task, and require rerunning compression per task. Moreover, compression after merging may destroy some of the fine-tuned signal.

These computational challenges hinder the accessibility of LLMs. An ideal system would enable knowledge transfer that is computationally inexpensive, does not require access to the original data, and retains the inherent advantages of adapters. We ask the research question below:

> *(Q) How to reuse the existing adapter synthesized from the original fine-tuning run of a base LLM and apply it to a compressed version of the base LLM?*

To address this question, we propose a new framework to seamlessly transfer personalized knowledge between a base LLM and its compressed version in a training-free and data-free scheme. This framework enables the ability to fine-tune once and deploy everywhere.

Our approach can help practitioners who may have a base model and many compressed models that require personalization, which would previously necessitate fine-tuning on every single model version. We provide a detailed discussion on several real-world use cases in Appendix C.

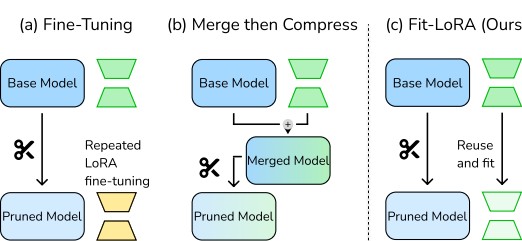

Figure 1: Possible workflows to personalize and compress LLMs. In a typical workflow (a), repeated fine-tuning is necessary. In (b), the resulting model becomes task-specific. This is undesirable, since each new task would require merging and running compression again, and storing the entire model per task. In (c), our training-free and data-free framework is the most efficient and maintains LoRA's modularity (see our second contribution bullet).

Our work specifically focuses on leveraging LoRA modules from the original fine-tuned model and apply them to a downstream compressed model, which is obtained by pruning the original base pretrained model. We show that our framework offers a solution to avoid a costly fine-tuning process. Our contributions are threefold:

1. We introduce FIT-LORA, a framework to transfer personalized knowledge between LLM scales. To the best of our knowledge, we are the first to show that it is possible to reuse LoRA adapters across different LLM scales, where the compressed LLM can be obtained via a variety of pruning techniques. Our method's ease of use, training-free and data-free characteristics, offers a valuable and practical way to achieve comparable task-specific performance on a variety of compressed models compared with direct task-specific fine-tuning.

2. Our method does not introduce any new constraints on LoRA fine-tuning or existing modern pruning techniques and is computationally instant, which increases practicality and likelihood of adoption. This plug-and-play experience enables the large corpus of existing LoRA adapters hosted on platforms such as HuggingFace to be directly used with our method, incentivizing the creation of new shareable adapters and increasing the value of already existing shared adapters from methods such as LoRAHub (Huang et al., 2024) and LoRA Land (Zhao et al., 2024).

3. We conducted extensive experiments across a wide range of downstream tasks, such as *BoolQ*, *SST-2*, *MRPC*, *RTE*, and *WinoGrande*. We validated our claims on multiple modern LLMs such as Llama-2-7B, Llama-3.1-8B, Mistral-7B, and Gemma-2-9B. To further demonstrate the generalizability of our framework, we demonstrate its applicability to multiple types of state-of-the-art pruning methods, including depth pruning, structured pruning, and sparsification.

## 2 RELATED WORK

**Parameter Efficient Fine-Tuning (PEFT).** Fine-tuning is necessary to adapt LLMs to downstream tasks. However, as LLMs grow larger and more complex, fine-tuning can be prohibitively expensive and time-consuming, requiring access to large amounts of memory and compute. PEFT

strategies aim to significantly reduce trainable parameters, thereby reducing the computational demands of fine-tuning. Several PEFT methods exist such as adapter tuning (Houlsby et al., 2019), prompt-tuning (Lester et al., 2021), and partial fine-tuning with masks (Guo et al., 2021). The most widely adopted PEFT methods follow the low-rank reparameterization paradigm such as LoRA (Hu et al., 2022) and other LoRA derivatives including QLoRA, VeRA, and DoRA (Dettmers et al., 2023; Kopiczko et al., 2024; Liu et al., 2024). In this paper, we focus on the transferability of LoRA.

**LLM Compression.** As LLMs continue to scale, there has been great interest in post-training LLM compression to enable flexible deployment. The two main research areas comprising LLM compression are quantization and pruning. In this paper, we focus on the more challenging task of pruning that introduces architectural changes to the pretrained LLM, which makes LoRA adapter transferability nontrivial. We discuss pruning in detail in Appendix A.

**LoRA Adapter Transfer.** Recently, there has been a variety of works that have explored the idea of transferring LoRA adapters between neural networks. Among the first, X-Adapter (Ran et al., 2024), proposes a universal mapper to transfer adapters. However, X-Adapter requires training for each target model and access to a shared data subset between the source and target model. LoRA-X (Farhadzadeh et al., 2025a) proposes specialized adapters that are transferrable by constraining the LoRA to be within the subspace of the source model weights. LoRA-X adapters only trains the singular values, which may hinder expressivity compared to vanilla LoRA adapters. ProLoRA (Farhadzadeh et al., 2025b) tackles this problem by decomposing the trained vanilla LoRA adapter into subspace and nullspace components and projecting it to the target model. However, all three focus on applying their method to diffusion models and not large scale language models.

Trans-LoRA (Wang et al., 2024) is able to transfer LoRAs across different LLM model families, but requires a small subset of the original fine-tuning dataset as well as a training procedure for each transfer with synthetic data generation. PortLLM (Shahroz et al., 2025) shows that LoRA adapters can be transferred as LLMs evolve. However, PortLLM (Shahroz et al., 2025) studies only model evolution, i.e., continual pretraining and does not consider LLM architecture change. LoRASuite (Li et al., 2025) proposes a LoRA transfer procedure across LLM upgrades, although it requires an additional lightweight fine-tuning step and access to a small amount of the original training data for good performance. Cross-LoRA (Xia et al., 2025) transfers adapters by aligning the source and target base model weight subspaces with SVD.

**Pre- and Post-Training Weight Transfer.** Xu et al. (2024) introduce a recipe for initializing smaller ViT models from a large ViT model via weight transfer. It consists of first-N mapping for layers, one-to-one mapping for components, and uniform mapping for elements. While our approach shares the spirit of weight selection as in Xu et al. (2024), instead of pretraining model initialization, our goal is to enable the immediate use of the smaller model without additional training.

Post-training weight transfer between LLMs to transfer instruction tuning, supervised-fine-tuning, and fine-tuning knowledge has been explored in Param$\Delta$ (Cao et al., 2025). As aforementioned, PortLLM (Shahroz et al., 2025) also shares the goal of fine-tuning knowledge transfer. In both works, the authors propose to use the difference in weights between the base and post-trained models and apply the task adapter to a new model checkpoint. However, both apply their methods to homologous models only, where there is no architectural change in the models.

**LoRA Merging.** Reusing LoRA adapters has garnered a lot of interest to boost performance on seen and unseen tasks. In LoRA Merging (Huang et al., 2024; Zhao et al., 2025), multiple task-specific LoRAs are composed to create a single adapter to address the problem of cross-task generalization. LoRAHub (Huang et al., 2024) proposes to linearly combine candidate LoRAs and applies a gradient-free step to tune the combination weights. LoRA-LEGO (Zhao et al., 2025) applies a rank-wise clustering heuristic into the merging process. Instead of cross-task generalization, our work focuses on cross-scale generalization.

Our proposed method, FIT-LoRA, aims to tackle a different problem than prior works by focusing on transferring LoRAs from a base LLM to its compressed versions. Additionally, we demonstrate that our method is effective (i) without needing potentially expensive SVD computations (Farhadzadeh et al., 2025a;b; Li et al., 2025; Xia et al., 2025), or (ii) fine-tuning training and (iii) access to the original fine-tuning dataset (Ran et al., 2024; Wang et al., 2024; Li et al., 2025).

## 3 PROPOSED METHOD

### 3.1 PRELIMINARIES

**Low-Rank Adaptation.** Consider a pretrained weight matrix $W_0 \in \mathbb{R}^{d \times k}$. Traditionally, fine-tuning would require computing gradients and storing optimizer states for the entire weight matrix $W_0$. LoRA (Hu et al., 2022) reparameterizes the weight update as a low-rank decomposition $W_{\text{new}} = W_0 + BA$, where $B \in \mathbb{R}^{d \times r}$ and $A \in \mathbb{R}^{r \times k}$, and the rank $r \ll \min(d, k)$. LoRA trains a fraction of parameters compared to full fine-tuning since only $B$ and $A$ are trainable and $W_0$ is frozen. LoRA adapters $\Delta W = BA$ can be kept separate (i.e., unmerged) from the pretrained LLM and stored and shared easily. Typically, one key limitation with LoRA adapters is that they can only be applied to the original LLM. Modifications made to the original LLM render already trained adapters obsolete.

**LLM-Streamline.** Depth pruning seeks to remove entire transformer layers from an LLM. LLM-Streamline (Chen et al., 2025) involves two stages: layer pruning and layer replacement. Contiguous $n$ layers are removed depending on the desired target sparsity ratio. The cosine similarity is computed against the input and output hidden states of the group of $n$ layers. Next, a single transformer layer is trained to estimate the $n$ layers that were removed. This is trained using an MSE loss using the input and output hidden states from the full LLM as training supervision.

**ReplaceME.** Similarly, ReplaceME (Shopkhoev et al., 2025) utilizes cosine similarity to determine which group of layers to prune. ReplaceME proposes a more efficient healing procedure to compensate for the lost layers, where a single linear transformation matrix is merged with the second feed-forward layer right before the deletion point. This transformation matrix is estimated using a numerical approach with the cosine distance objective and an ADAM optimizer, again using the input and output hidden states.

**Olica.** Olica (He & Lin, 2025) is a structured pruning method that prunes the multi-head attention (MHA) and feed-forward network (FFN) layers in an LLM and claims to achieve better performance than previous works. The key insight in Olica is that when pruning the MHA layer, the matrix products $W_q W_k^\top$ and $W_v W_o^\top$ should be treated as unified entries. Instead of applying SVD separately to $W_v$ and $W_o$, SVD is applied to the product $W_{vo} = W_v W_o^\top$ which leads to better reconstruction. After this decomposition, certain columns corresponding to a group of parameters of a neuron are deleted utilizing the LoRAP method (Li et al., 2024).

**MaskLLM.** MaskLLM (Fang et al., 2024) introduces learnable masks, while addressing the challenge of combinatorial and non-differentiable nature of selecting an optimal masks. MaskLLM is able to learn high-quality masks by differentiable sampling via the Gumbel softmax. Since MaskLLM claims to outperform previous works such as SparseGPT and Wanda, while offering value through inference speedups, we choose to evaluate our proposed method FIT-LORA on it.

### 3.2 PROPOSED CROSS-SCALE ADAPTATION WITH FIT-LORA

Our framework reuses the original LoRA adapters from the base model while fitting within the form factor of the pruned LLM as seen in Figure 2. Typically, there can be LoRA $BA$ modules for up to seven matrices per transformer layer. Four in the MHA layer: $W_q$, $W_k$, $W_v$, $W_o$, and three in the MLP layer: $W_{\text{gate}}$, $W_{\text{up\_proj}}$, $W_{\text{down\_proj}}$. We copy and trim these modules when necessary.

**Depth Mismatch.** Depth pruning removes entire transformer layers from the LLM. Consider the Llama-2-7B model with 32 layers. A pruned model obtained from either LLM-Streamline (Chen et al., 2025) or ReplaceME (Shopkhoev et al., 2025) may prune layers 24 to 29, resulting in a smaller Llama-2 model with 26 layers. When transferring the original task adapter to the 26-layer model, FIT-LORA discards the LoRA modules corresponding to layers 24–29 and preserves the rest.

**Dimension Mismatch.** Structured pruning can modify the width or height of dense weight matrices in the MHA or MLP layers. Consider the Llama-2-7B model with value and output projection matrices given by $W_v \in \mathbb{R}^{4096 \times 4096}$ and $W_o \in \mathbb{R}^{4096 \times 4096}$. Structured pruning via Olica (He &

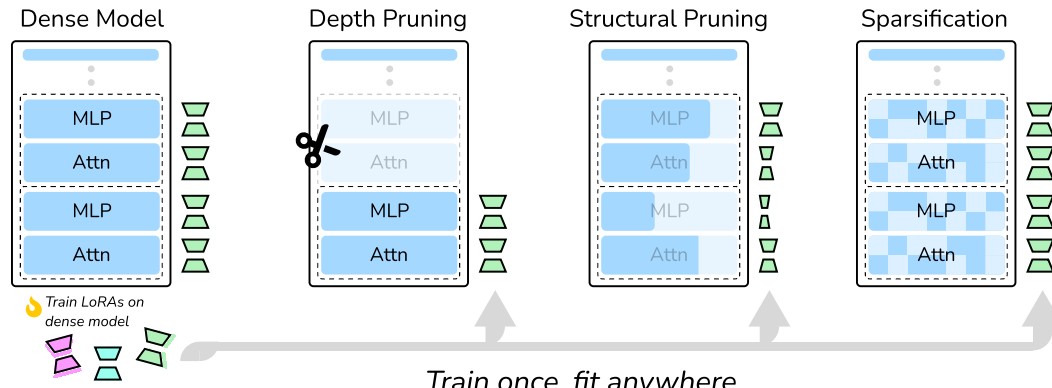

Figure 2: Model services need a collection of different sized models for diverse deployment scenarios. Often, different types of pruning methods are desired to get compressed models. Personalization for each model is needed for enhanced performance on downstream tasks. Our framework adds personalization to each model in the collection by leveraging the original LoRA adapter, avoiding the need for repeated and costly fine-tuning.

Lin, 2025) prunes certain columns and rows of these matrices that results in a dimension reduction where $W_v \in \mathbb{R}^{3712 \times 4096}$ and $W_o \in \mathbb{R}^{4096 \times 3712}$. In FIT-LORA, we fit the LoRA modules to the new dimension by removing rows or columns that correspond to the pruned row and column indexes. We reduce $B_v \in \mathbb{R}^{4096 \times 8} \to B_v \in \mathbb{R}^{3712 \times 8}$ and preserve $A_v \in \mathbb{R}^{8 \times 4096}$ yielding $W_{v\_new} = W_v + B_v A_v$. Similarly, we reduce $A_o \in \mathbb{R}^{8 \times 4096} \to A_o \in \mathbb{R}^{8 \times 3712}$ and preserve $B_o \in \mathbb{R}^{4096 \times 8}$ yielding $W_{o\_new} = W_o + B_o A_o$. Olica also prunes the MLP layers which affects $W_{gate}, W_{up\_proj}, W_{down\_proj}$. FIT-LORA applies the same procedure as described above to fit the LoRA modules to the new pruned dimensions.

**Sparsification.** Unstructured pruning and semi-structured pruning result in sparse models where weights are zeroed out. LoRA adapters can be directly transferred to the sparse model in this scenario. This is because LoRA modules are inherently separate from the base LLM. While the base LLM can remain sparse to enjoy the benefits of potential memory and computational gains, the task-specific LoRA can be dense. Although the full-precision LoRA adapter reintroduces some overhead, it does not offset all the savings from the sparse model. The sparse model can yield memory savings and the largest `matmuls` can still take advantage of `spmv` operations, while the full-precision LoRA adapter `matmuls` are small due to its low-rank structure.

Augmenting sparse models with low-rank approximation paths has been studied in works such as EoRA (Liu et al., 2025) and SLiM (Mozaffari et al., 2025). In FIT-LORA, we investigate whether such addition of low-rank paths using SLiM-LoRA from Mozaffari et al. (2025) can also result in better transferability of LoRA adapters on downstream tasks.

**Why Does FIT-LORA Work?** The design of our approach exploits the fact that the pruned model's layers are aligned with those of the base model. This is true in both examples of LLM-Streamline and ReplaceME, where the pruning algorithm keeps the surviving layer weights unchanged while incorporating a healing block. The healing block is trained to reconstruct the hidden outputs by the group of eliminated layers, ensuring that the surviving layers continue to receive the same input regardless from the pruned layers or the healing block. Since the weights of the surviving layers in the compressed models are unchanged, applying LoRA updates to the surviving layers can preserve most of the task adaptation signal. The only mismatch concerns the LoRA updates from the eliminated layers, which are skipped for the healing block. Empirically, we find that this copy-and-trim strategy is remarkably effective, revealing that task-specific LoRA knowledge is robust to architectural reduction.

We note that the structural correspondence between the pruned and base model is crucial for LoRA adapter transferability. Our approach in transferring adapters may not be suitable for compression

Table 1: Zero-shot performance comparison of FIT-LORA and baseline models with Llama-2-7B architecture and **LLM-Streamline** (Chen et al., 2025) as pruning method. Pruned LLM has **10** layers removed and is 4.7B parameters.

| Model Version | BoolQ
Accuracy | SST-2
Accuracy | MRPC
Accuracy/F1 | RTE
Accuracy | WinoGrande
Accuracy |
|---|---|---|---|---|---|
| Pretrained LLM | 77.68 | 49.43 | 69.12 / 81.50 | 62.82 | 68.90 |
| Pretrained LLM w/ LoRA FT | 89.02 | 96.33 | 89.71 / 92.45 | 88.45 | 75.77 |
| Pruned LLM | 74.70 | 51.61 | 69.10 / 81.12 | 61.70 | 66.69 |
| Pruned LLM w/ LoRA FT | 89.08 | 96.33 | 89.95 / 92.64 | 88.81 | 71.98 |
| Pruned LLM w/ FIT-LORA | 88.99 | 95.99 | 89.71 / 92.50 | 88.09 | 79.48 |

algorithms that do not follow this assumption. For instance, pruning methods that disrupt layerwise independence would generally not be conducive for LoRA transferability.

## 4 EXPERIMENTS

### 4.1 EXPERIMENTAL SETTINGS

**Datasets and Models.** We evaluate our FIT-LORA framework on a diverse set of datasets to showcase its effectiveness across a variety of downstream tasks and domains. We utilize five popular evaluation datasets including *BoolQ* (Clark et al., 2019), *SST-2* (Wang et al., 2019b), *MRPC* (Wang et al., 2019b), *RTE* (Wang et al., 2019b), and *WinoGrande* (Sakaguchi et al., 2021). For question answering tasks, we use *BoolQ* from the *SuperGLUE* benchmark (Wang et al., 2019a) and *SST-2* from the *GLUE* benchmark (Wang et al., 2019b). For similarity and paraphrase tasks, we use *MRPC* from *GLUE*. For commonsense inference tasks, we use *RTE* from *GLUE*. For commonsense reasoning tasks, we use *WinoGrande*. We primarily leverage the Llama-2-7B model (Touvron et al., 2023) to evaluate our framework, as it is the most widely supported in previously released research codebases. In addition, we also evaluate FIT-LORA on other, more modern models such as Llama-3.1-8B-Instruct (Grattafiori et al., 2024), Mistral-7B-Instruct-v0.3 (Jiang et al., 2023), and Gemma-2-9B-Instruct (Team et al., 2024), to demonstrate the robustness of the FIT-LORA framework towards different model architectures.

**Training, Pruning, and Evaluation.** To obtain the task-specific LoRA adapters, we undergo the usual LoRA fine-tuning process on the task dataset and train the MHA layer: q_proj, k_proj, v_proj, o_proj, and the MLP layer: up_proj, down_proj, gate_proj where applicable. The LoRA rank is set as $r = 8$, alpha as $\alpha = 16$, with a learning rate of $0.001$ for 5 epochs. These hyperparameters are consistent across all our experiments. To obtain the pruned models, we leverage the official Llama-2 pruned checkpoints from LLM-Streamline and ReplaceME. Additionally, to acquire more pruned models such as pruned Mistral, we run the ReplaceME code with the following configuration: one linear transformation as replacement, optimized with the cosine objective function, and SlimOrca as the calibration dataset. Similarly, for MaskLLM, we leverage their checkpoints, and we use the recommended and default configurations for Olica pruning. For SLiM-LoRA, we use MaskLLM as initialization and run the default configuration to obtain the low-rank paths. The evaluation suite we utilize to measure the performance of pretrained, fine-tuned, and FIT-LORA models is the Language Model Evaluation Harness `lm-eval` (Gao et al., 2024) with the *zero-shot* setting and *accuracy* metric. Our implementation builds on the HuggingFace `transformers` and `peft` libraries as well as the released repositories from previous pruning methods. Our experiments were primarily conducted on 1 NVIDIA GH200 GPU (96GB).

### 4.2 MAIN RESULTS

We first present the main results of our framework FIT-LORA against several key baselines. In our experiments, we compare directly training a LoRA adapter against our training-free framework. When comparing the Pretrained LLM with (w/) LoRA Fine-Tuning (FT) vs transferring this trained adapter to the Pruned LLM with FIT-LORA, we find that our framework is able to preserve the

Table 2: Zero-shot performance comparison of FIT-LORA and baseline models with Llama-2-7B architecture and **ReplaceME** (Shopkhoev et al., 2025) as pruning method. Pruned LLM has 8 layers removed and is 5B parameters.

| Model Version | BoolQ Accuracy | SST-2 Accuracy | MRPC Accuracy/F1 | RTE Accuracy | WinoGrande Accuracy |
|---|---|---|---|---|---|
| Pretrained LLM | 77.68 | 49.43 | 69.12 / 81.50 | 62.82 | 68.90 |
| Pretrained LLM w/ LoRA FT | 89.02 | 96.33 | 89.71 / 92.45 | 88.45 | 75.77 |
| Pruned LLM | 71.87 | 56.54 | 68.39 / 81.22 | 57.76 | 66.38 |
| Pruned LLM w/ LoRA FT | 89.42 | 96.33 | 88.48 / 91.77 | 87.36 | 79.87 |
| Pruned LLM w/ FIT-LORA | 88.69 | 96.44 | 88.97 / 92.09 | 88.09 | 79.40 |

Table 3: Performance gains using FIT-LORA on a depth pruned model with **ReplaceME** (Shopkhoev et al., 2025). Pruned LLM has 8 layers removed.

| Model | BoolQ Accuracy | SST-2 Accuracy | MRPC Accuracy | RTE Accuracy | WinoGrande Accuracy |
|---|---|---|---|---|---|
| Llama-2 7B | 77.68 | 49.43 | 69.12 | 62.82 | 68.90 |
| Pruned Llama-2 5B | 71.87 | 56.54 | 68.39 | 57.76 | 66.38 |
| Pruned Llama-2 5B w/ FIT-LORA | 88.69 (↑ **16.82**) | 96.44 (↑ **39.90**) | 88.97 (↑ **20.58**) | 88.09 (↑ **30.33**) | 79.40 (↑ **13.02**) |

task-specific knowledge even under significant LLM architecture change such as removing entire transformer layers. In Table 1, we see that pruned LLM with FIT-LORA is maximally within $-0.36$ {*RTE*} of the pretrained LLM with LoRA fine-tuning. Remarkably, in some cases such as on the *WinoGrande* dataset, our framework even outperforms the pretrained fine-tuned model by $+3.71$. When comparing the Pruned LLM with direct LoRA fine-tuning vs our method, we see similar results where the transfer framework matches the performance compared to conducting expensive fine-tuning.

Table 3 emphasizes the role of FIT-LORA in enabling strong task-specific knowledge transfer between LLMs. We observe that across all five tasks {*BoolQ, SST-2, MRPC, RTE, WinoGrande*}, FIT-LORA achieves significant performance gains over the Pruned LLM baseline {$+16.82$, $+39.90$, $+20.58$, $+30.33$, $+13.02$}. Importantly, we note that the Pruned LLM baselines perform weaker on downstream tasks due to the lossy nature of pruning LLMs. For instance, in Table 3, the Pruned Llama-2 5B model has a $-6$ regression on *BoolQ* and a $-5$ regression on *RTE*. Performance degradations can be exacerbated depending on the pruning method, such as for 50% sparsity in Table 5. This further necessitates the need for fine-tuning to adapt pruned LLMs to downstream tasks and motivates the need for our training and data-free framework.

**Consistency Across Pruning Methods.** We demonstrate our framework's versatility and usefulness by evaluating its performance on multiple pruning methods. In Tables 1 and 2, we analyze the LoRA transfer for depth pruning on LLM-Streamline and ReplaceME. In both scenarios, downstream task performance transfer on all five tasks has the same effect as conducting direct LoRA fine-tuning. This strong result implies that the LoRA weight updates associated with the pruned layers also do not contribute downstream critical information. Table 2 shows our results under the structured pruning scenario with Olica. We highlight that FIT-LORA can perform comparably to direct LoRA fine-tuning. We note that although, the *RTE* accuracy with the transferred adapter has a performance gap against direct fine-tuning, the overall performance gain delta $+26.36$ between pruned LLM (55.23) to pruned LLM with FIT-LORA (81.59) is actually greater than that of the delta $+25.63$ between the pretrained LLM (62.82) to pretrained LLM with LoRA fine-tuning (88.45). The demonstrates the synergy between our framework and existing pruning works. Naturally, if the pruning baseline is weaker, the transferred performance will also be impacted. We explore this notion in Section 4.3 no healing.

In Table 5, we report the performance of our framework under unstructured pruning, i.e., sparsification with the 2:4 pattern with MaskLLM. This experiment also showcases FIT-LORA's resilience to high 50% sparsity ratios. While the sparse LLM baseline has a sizeable performance drop {$-6.88, -17.39/-19.70, -5.78$} across {*BoolQ, MRPC, RTE*} compared to the pretrained

Table 4: Zero-shot performance comparison of FIT-LoRA and baseline models with Llama-2-7B architecture and **Olica** (He & Lin, 2025) as pruning method. Pruned LLM is 5.4B parameters.

| Model Version | BoolQ Accuracy | SST-2 Accuracy | MRPC Accuracy/F1 | RTE Accuracy | WinoGrande Accuracy |
|---|---|---|---|---|---|
| Pretrained LLM | 77.68 | 49.43 | 69.12 / 81.50 | 62.82 | 68.90 |
| Pretrained LLM w/ LoRA FT | 89.02 | 96.33 | 89.71 / 92.45 | 88.45 | 75.77 |
| Pruned LLM | 72.48 | 58.83 | 69.36 / 81.32 | 55.23 | 67.40 |
| Pruned LLM w/ FIT-LoRA | 85.41 | 95.87 | 86.27 / 90.14 | 81.59 | 78.77 |

Table 5: Zero-shot performance comparison of FIT-LoRA and baseline models with Llama-2-7B architecture and **MaskLLM** (Fang et al., 2024) as pruning method. Sparse LLM has 50% sparsity with 2:4 pattern.

| Model Version | BoolQ Accuracy | SST-2 Accuracy | MRPC Accuracy/F1 | RTE Accuracy | WinoGrande Accuracy |
|---|---|---|---|---|---|
| Pretrained LLM | 77.68 | 49.43 | 69.12 / 81.50 | 62.82 | 68.90 |
| Pretrained LLM w/ LoRA FT | 89.02 | 96.33 | 89.71 / 92.45 | 88.45 | 75.77 |
| Sparse LLM | 70.80 | 52.52 | 51.23 / 61.80 | 57.04 | 65.27 |
| Sparse LLM + SLiM-LoRA | 76.33 | 49.08 | 66.18 / 78.90 | 54.15 | 66.93 |
| Sparse LLM w/ LoRA FT | 87.58 | 95.87 | 87.01 / 90.59 | 87.73 | 76.32 |
| Sparse LLM w/ FIT-LoRA | 85.27 | 95.76 | 87.01 / 90.75 | 81.23 | 68.98 |
| Sparse LLM + SLiM-LoRA w/ FIT-LoRA | 86.73 | 96.22 | 87.99 / 91.51 | 84.48 | 71.67 |

LLM, FIT-LoRA successfully recovers the task-specific performance making it on par with the pretrained fine-tuned model. When incorporating compensation techniques for sparse models with SLiM-LoRA, we get additional performance benefits across all tasks. This finding further highlights the versatility of our framework and its ability to complement other research techniques.

**Consistency Across LLM Architectures.** We evaluate our proposed method on different LLM architectures in Table 6. Across all five tasks, and each LLM architecture, we find that our transfer framework consistently performs comparably to the original pretrained backbone with fine-tuning, while operating on a smaller and more efficient pruned backbone. In some cases, we observe that our framework even exceeds the performance of the original fine-tuned model. This can seen in Llama-3.1-8B-Instruct *BoolQ* +2.45, in Mistral-7B-Instruct-v0.3 *BoolQ* +0.30, and in Gemma-2-9B-Instruct *MRPC* +0.49/+0.32. It is possible that directly fine-tuning on the pruned model can cause some performance drop compared to the original fine-tuned base model, due to different training dynamics associated with the smaller model. For example, in Mistral-7B-Instruct-v0.3 *MRPC*, there is a $-5.89/-3.26$ drop from the 2nd row to the 4th row. Notably, FIT-LoRA in the 5th row, is able to outperform even pruned model direct fine-tuning, showing the potential of distilling knowledge learned by a larger model to a smaller model. Our results from Table 6 give us additional confidence that FIT-LoRA can be adopted regardless of the model family used.

## 4.3 ADDITIONAL EXPERIMENTS

We also report additional experiments related to the sensitivity of FIT-LoRA under different settings. We conduct experiments on LoRA rank and LoRA derivatives: QLoRA and DoRA in Appendix B.

**Model Evolution.** As studied in PortLLM (Shahroz et al., 2025), LoRA adapters can be successfully reused as models undergo periodic updates via continual pretraining. We study whether this two-step modification, i.e., model evolution followed by pruning, can still leverage the original adapters. In Table 7, we run continual pretraining with the SlimOrca dataset and then use depth pruning. In all cases, the performance is comparable to only pruning, and in three out of five datasets {*BoolQ*, *SST-2*, *WinoGrande*}, the performance increases, showing that continual pretraining knowledge can aid in downstream tasks.

Table 6: Zero-shot performance comparison of FIT-LoRA and baseline models with multiple different architectures: Llama-2-7B, Llama-3.1-8B, Mistral-7B, Gemma-2-9B and **ReplaceME** (Shopkhoev et al., 2025) as pruning method. Each pruned LLM has 8 layers removed.

| Model Architecture | Model Version | BoolQ Accuracy | SST-2 Accuracy | MRPC Accuracy/F1 | RTE Accuracy | WinoGrande Accuracy |
|---|---|---|---|---|---|---|
| Llama-2-7B | Pretrained LLM | 77.68 | 49.43 | 69.12 / 81.50 | 62.82 | 68.90 |
| | Pretrained LLM w/ LoRA FT | 89.02 | 96.33 | 89.71 / 92.45 | 88.45 | 75.77 |
| | Pruned LLM | 71.87 | 56.54 | 68.39 / 81.22 | 57.76 | 66.38 |
| | Pruned LLM w/ LoRA FT | 89.42 | 96.33 | 88.48 / 91.77 | 87.36 | 79.87 |
| | Pruned LLM w/ FIT-LoRA | 88.69 | 96.44 | 88.97 / 92.09 | 88.09 | 79.40 |
| Llama-3.1-8B-Instruct | Pretrained LLM | 82.87 | 89.22 | 71.32 / 82.14 | 75.00 | 73.72 |
| | Pretrained LLM w/ LoRA FT | 88.99 | 95.99 | 87.50 / 91.49 | 89.89 | 75.85 |
| | Pruned LLM | 82.57 | 90.14 | 69.12 / 81.36 | 68.59 | 71.98 |
| | Pruned LLM w/ LoRA FT | 88.13 | 95.99 | 86.27 / 90.67 | 91.70 | 80.35 |
| | Pruned LLM w/ FIT-LoRA | 91.44 | 95.76 | 89.46 / 92.55 | 90.97 | 80.51 |
| Mistral-7B-Instruct-v0.3 | Pretrained LLM | 86.85 | 81.65 | 71.08 / 82.28 | 73.65 | 74.19 |
| | Pretrained LLM w/ LoRA FT | 88.69 | 95.41 | 89.71 / 92.58 | 90.25 | 70.40 |
| | Pruned LLM | 73.09 | 88.07 | 68.38 / 81.22 | 75.45 | 66.61 |
| | Pruned LLM w/ LoRA FT | 89.66 | 94.95 | 83.82 / 89.32 | 91.34 | 76.64 |
| | Pruned LLM w/ FIT-LoRA | 88.99 | 95.64 | 89.46 / 92.42 | 89.89 | 80.51 |
| Gemma-2-9B-Instruct | Pretrained LLM | 89.60 | 92.09 | 74.02 / 81.97 | 76.53 | 75.85 |
| | Pretrained LLM w/ LoRA FT | 92.97 | 96.22 | 89.46 / 92.50 | 93.50 | 83.19 |
| | Pruned LLM | 86.85 | 91.97 | 75.51 / 83.90 | 77.26 | 73.40 |
| | Pruned LLM w/ LoRA FT | 91.41 | 96.67 | 88.97 / 92.17 | 93.86 | 83.19 |
| | Pruned LLM w/ FIT-LoRA | 92.97 | 96.44 | 89.95 / 92.82 | 93.50 | 80.82 |

Table 7: Two-stage evaluation with model evolution (continual pretraining) and model pruning with Llama-2-7B architecture and **ReplaceME** (Shopkhoev et al., 2025) as pruning method. First continual pretraining is done on the base model using SlimOrca dataset, then it is pruned. Pruned LLM has 8 layers removed and is 5B parameters.

| Model Version | BoolQ Accuracy | SST-2 Accuracy | MRPC Accuracy/F1 | RTE Accuracy | WinoGrande Accuracy |
|---|---|---|---|---|---|
| Pruned LLM w/ FIT-LoRA | 88.69 | 96.44 | 88.97 / 92.09 | 88.09 | 79.40 |
| Model Evolution + Pruned LLM w/ FIT-LoRA | 89.08 | 96.67 | 88.48 / 91.74 | 87.36 | 80.27 |

Table 8: Performance comparison of no healing pruning with Llama-2-7B architecture and **ReplaceME** (Shopkhoev et al., 2025) as pruning method. Pruned LLM has 8 layers removed and is 5B parameters.

| Model Version | BoolQ Accuracy | SST-2 Accuracy | MRPC Accuracy/F1 | RTE Accuracy | WinoGrande Accuracy |
|---|---|---|---|---|---|
| Pruned LLM No Healing w/ FIT-LoRA | 50.43 | 96.33 | 87.25 / 90.37 | 47.29 | 76.56 |
| Pruned LLM w/ FIT-LoRA | 88.69 | 96.44 | 88.97 / 92.09 | 88.09 | 79.40 |

**No Healing.** We conduct an ablation-inspired analysis to explore how variations in pruning could impact our framework's performance. LLM pruning often involves a healing procedure to compensate for deleting parts of the network. In ReplaceME, a learned linear transformed is merged into part of the network. In Table 8, we find that the transfer performance can be impacted significantly when the healing step is omitted. This observation is in line with pruning literature where healing is necessary to get a reasonable baseline model. We find that the transferability of adapters is correlated with the quality of pruning.

**Scalability.** In Table 9, we investigate whether our framework can maintain its performance at larger scales. Specifically, we evaluate on Llama-2 13B which has 40 layers. We keep the same pruning ratio at 25% and target 10 layers for removal with ReplaceME. Our framework is able to significantly improve performance on each dataset compared to the pruned model. On average we see an increase of +23.64 for Llama-2 9.8B vs +22.58 for Llama-2 5B. This reinforces our framework's wide applicability to LLMs and can imply that larger scale models can have a larger delta improvement.

Table 9: Scalability: Performance gains using FIT-LORA on Llama-2-13B a with **ReplaceME** (Shopkhoev et al., 2025). Pruned (✂) Llama-2 9.8B LLM has 10 layers removed. Pruned Llama-2 5B has 8 layers removed.

| Model | BoolQ Accuracy | MRPC Accuracy | RTE Accuracy | Average Accuracy |
|---|---|---|---|---|
| Llama-2 7B | 77.68 | 69.12 | 62.82 | 69.87 |
| Llama-2 7B w/ LoRA FT | 89.45 | 90.20 | 89.89 | 89.85 |
| ✂ Llama-2 5B | 71.87 | 68.39 | 57.76 | 66.00 |
| ✂ Llama-2 5B w/ FIT-LORA | 88.69 (↑ **16.82**) | 88.97 (↑ **20.58**) | 88.09 (↑ **30.33**) | 88.58 (↑ **22.58**) |
| Llama-2 13B | 80.52 | 68.63 | 65.34 | 71.49 |
| Llama-2 13B w/ LoRA FT | 89.45 | 90.20 | 89.89 | 89.84 |
| ✂ Llama-2 9.8B | 62.14 | 68.38 | 67.15 | 65.89 |
| ✂ Llama-2 9.8B w/ FIT-LORA | 88.13 (↑ **25.19**) | 90.93 (↑ **22.55**) | 89.53 (↑ **22.38**) | 89.53 (↑ **23.64**) |

## 5 CONCLUSION

To the best of our knowledge, this is the first investigation that aims to transfer personalized knowledge between base and pruned LLMs. We propose FIT-LORA, a training-free and data-free framework to address this challenge. We show the promise of our framework by achieving results comparable to direct fine-tuning on a diverse set of tasks, LLM backbones, and across multiple types of modern pruning techniques.

**Limitations.** Currently, FIT-LORA addresses the transferability challenge between base model to pruned model. However, future research could extend this to address the challenge of transferability between entirely heterogeneous models, such as from Llama to Gemma.

## ETHICS STATEMENT

We adhere to the ICLR Code of Ethics. No private, sensitive, or personally identifiable data are involved. Our work does not raise foreseeable ethical concerns or produce harmful societal outcomes.

## REPRODUCIBILITY STATEMENT

Our goal is to ensure that our results are reproducible and that anyone may implement and build on our approach. In the paper, we outline detailed experimental settings in subsection 4.1. We also plan to release the source code and models.

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

## A   MORE RELATED WORKS

Pruning methods aim to identify and remove redundant parameters from neural networks. LLM pruning methods fall into three main categories: unstructured, structured, and depth pruning.

**Unstructured Pruning.**   Unstructured pruning introduce sparsity in the model by zeroing out certain weights in the weight matrices. SparseGPT (Frantar & Alistarh, 2023) solves a layer-wise reconstruction problem using calibration data and employs an efficient weight update procedure via Hessians to minimize pruning accuracy drop. Wanda (Sun et al., 2024) utilizes weight magnitudes and input activate norms to remove weights without the need for weight updates. Although, SparseGPT and Wanda (Frantar & Alistarh, 2023; Sun et al., 2024) perform well under unstructured sparsity, they require specialized hardware to get practical computation speedups with the resulting sparse model. A focus on hardware friendly semi-structured patterns such as NVIDIA's 2:4 sparsity pattern (Mishra et al., 2021) is prevalent with works like MaskLLM (Fang et al., 2024). MaskLLM (Fang et al., 2024) introduces a learnable mask trained on a large corpus to achieve better performance for 2:4 sparsity patterns.

**Structured Pruning.**   Structured pruning on the other hand, remove entire structured components of the network. This may involve eliminating attention heads or MLP channels of the network. Pioneering works such as LLM-Pruner (Ma et al., 2023) groups neurons across modules based on their dependencies and eliminates the entire group based on its overall importance. Structured pruning by applying SVD based approaches on dense matrices have also gained popularity through works such as SVD-LLM, LoRAP, and Dobi-SVD (Wang et al., 2025; Li et al., 2024; Qinsi et al., 2025). Recently, Olica (He & Lin, 2025) has demonstrated strong performance without needing access to lots of calibration data.

**Depth Pruning.**   Depth pruning, also known as layer pruning or layer skipping, has shown to be a promising approach for compressing LLMs by removing entire transformer layers, especially in the deeper layers (Gromov et al., 2025). Depth pruning may be preferred over other pruning methods since it keeps architecture uniformity and offers more straightforward GPU memory savings and computation speedups. Shortened llama (Kim et al., 2024) removes transformer layers from the LLM based on Taylor and PPL metrics with an optional LoRA recovery stage. Further improvements in depth pruning by removing consecutive layers and introducing a single lightweight layer replacement is demonstrated in LLM-Streamline (Chen et al., 2025). Recently, ReplaceME (Shopkhoev et al., 2025) introduces a training-free approach that removes layers from the LLM and compensates via a linear transformation estimation with a small calibration dataset.

## B   MORE EXPERIMENTS

**Applicability to other LoRA derivatives.**   The popularity of LoRA has led to many other LoRA variants being introduced. We evaluate to what extent other PEFT adapters can capture and transfer fine-tuned knowledge to pruned LLMs in Table 10, with a focus on QLoRA (Dettmers et al., 2023) and DoRA (Liu et al., 2024) as they are the most widely adopted variants. We observe that LoRA adapters can be better at *MRPC*, while QLoRA and DoRA adapters have an advantage in *RTE*. Our results show evidence that FIT-LORA is generalizable to other LoRA derivatives.

**Change in LoRA rank $r$.**   Users can adjust the LoRA rank $r$ during fine-tuning, which typically involves a trade-off between performance and efficiency. In Table 11, we show the impact of choosing a different initial $r$ on the transfer procedure. In all three datasets and all five ranks, downstream performance after transferring and fitting has no significant impact, and generally ranks between 4 and 16 are preferred. This reinforces our framework's durability under different rank selections, where different ranks can be chosen depending on task complexity.

Table 10: Performance comparison of different PEFT methods, LoRA (Hu et al., 2022), QLoRA (Dettmers et al., 2023), DoRA (Liu et al., 2024) on five compression methods, three tasks with Llama-2-7B. Each compression method uses the same pruning ratios as in the previous tables.

| PEFT | Model Version | Adaptation | BoolQ Accuracy | MRPC Accuracy/F1 | RTE Accuracy |
|------|---------------|------------|----------------|------------------|--------------|
| | Pretrained | – | 77.68 | 69.12 / 81.50 | 62.82 |
| LoRA | Pretrained | Fine-Tuned | 89.02 | 89.71 / 92.45 | 88.45 |
| | ReplaceME | FIT-LORA | 88.69 | 88.97 / 92.05 | 88.09 |
| | LLM-Streamline | FIT-LORA | 88.99 | 89.71 / 92.50 | 88.09 |
| | Olica | FIT-LORA | 85.41 | 86.27 / 90.14 | 81.59 |
| | MaskLLM | FIT-LORA | 85.27 | 87.01 / 90.75 | 81.23 |
| | MaskLLM + Slim-LoRA | FIT-LORA | 86.73 | 87.99 / 91.51 | 84.48 |
| QLoRA | Pretrained | Fine-Tuned | 88.93 | 88.97 / 91.92 | 89.89 |
| | ReplaceME | FIT-QLORA | 87.71 | 88.73 / 91.93 | 90.25 |
| | LLM-Streamline | FIT-QLORA | 88.87 | 89.46 / 92.36 | 89.17 |
| | Olica | FIT-QLORA | 85.69 | 83.33 / 88.67 | 84.48 |
| | MaskLLM | FIT-QLORA | 84.98 | 84.07 / 88.58 | 84.12 |
| | MaskLLM + Slim-LoRA | FIT-QLORA | 86.42 | 87.01 / 90.78 | 88.09 |
| DoRA | Pretrained | Fine-Tuned | 88.87 | 89.22 / 92.23 | 89.89 |
| | ReplaceME | FIT-DORA | 81.95 | 90.20 / 92.86 | 88.09 |
| | LLM-Streamline | FIT-DORA | 88.90 | 89.46 / 92.36 | 89.17 |
| | Olica | FIT-DORA | 85.63 | 82.84 / 88.37 | 83.39 |
| | MaskLLM | FIT-DORA | 85.23 | 83.33 / 88.44 | 84.48 |
| | MaskLLM + Slim-LoRA | FIT-DORA | 86.88 | 86.27 / 90.34 | 88.45 |

Table 11: Performance comparison of different LoRA ranks $(2, 4, 8, 16, 32)$ on three tasks with Llama-2-7B architecture and **ReplaceME** (Shopkhoev et al., 2025) as pruning method. Pruned LLM has 8 layers removed and is 5B parameters.

| Rank ($r$) | Model Version | BoolQ Accuracy | MRPC Accuracy/F1 | RTE Accuracy |
|------------|---------------|----------------|------------------|--------------|
| – | Pretrained LLM | 77.68 | 69.12 / 81.50 | 62.82 |
| $r = 2$ | Pretrained LLM w/ LoRA FT | 89.54 | 89.95 / 92.82 | 88.81 |
| | Pruned LLM w/ FIT-LORA | 89.24 | 87.99 / 91.54 | 88.45 |
| $r = 4$ | Pretrained LLM w/ LoRA FT | 89.08 | 91.42 / 93.81 | 89.53 |
| | Pruned LLM w/ FIT-LORA | 88.93 | 90.93 / 93.52 | 89.53 |
| $r = 8$ | Pretrained LLM w/ LoRA FT | 89.02 | 89.71 / 92.45 | 88.45 |
| | Pruned LLM w/ FIT-LORA | 88.69 | 88.97 / 92.05 | 88.09 |
| $r = 16$ | Pretrained LLM w/ LoRA FT | 89.69 | 90.44 / 93.02 | 88.81 |
| | Pruned LLM w/ FIT-LORA | 88.96 | 90.44 / 93.22 | 89.89 |
| $r = 32$ | Pretrained LLM w/ LoRA FT | 89.11 | 89.46 / 92.28 | 88.09 |
| | Pruned LLM w/ FIT-LORA | 88.75 | 88.97 / 92.15 | 88.09 |

## C    DISCUSSION ON REAL-WORLD USE CASES

We outline several concrete use cases that motivate a training-free and data-free framework:

**No data.**    Model fine-tuning requires task-specific training data that may not be available at all. Consider the following scenarios:

- A downstream developer wants to leverage a LoRA adapter that was distributed by a model provider or another platform. The original data used to train the adapter is not available. And of course, it would be unrealistic for a downstream developer to collect and annotate their own dataset that matches the task specification.

- Under privacy policies, it may be required that data collected be deleted after a certain amount of time. Data used for a previous fine-tuning run may not be available forever. There is also no "auxiliary public data" given that this type of data requires authorized access.

- The original data used for fine-tuning were seen only once and never stored, e.g., in an online fine-tuning scenario, especially under restrictive privacy constraints where the provider cannot store user-sensitive data.

**Can't fine-tune.**    Downstream developers may not have access to the GPU resources required for training. While an edge device may have the required memory for inference, it may not have enough memory for training, which is more expensive. For training, one has to store optimizer states and gradients in memory.

**Large cost, time and logistical savings.**    More challenging to keep up with model customization as the number of deployment scenarios grows. Consider that one needs to support 100s of compressed models that use different pruning techniques or have unique pruning ratios. One can also have many LoRA adapters, each for a different task. Fine-tuning each model on each task becomes cumbersome, time-consuming, and costly.

