# OpenReview forum: "Fit-LoRA: Fit Your LoRAs to Pruned LLMs Without Additional Training or Data"
_ICLR.cc/2026/Conference — Submitted to ICLR 2026_

### Official Review · Reviewer_ySHe · 2025-10-28

**Soundness:** 3
**Presentation:** 3
**Contribution:** 2
**Rating:** 4
**Confidence:** 3

**Summary:**

The paper introduces FIT-LoRA, a training-free and data-free framework that enables the reuse of LoRA adapters on pruned or compressed versions of large language models (LLMs). It supports depth pruning, structural pruning, and sparsification, effectively addressing the incompatibility issues that arise when a fine-tuned LoRA adapter is applied to a model that has been modified through pruning.

**Strengths:**

1. The paper is the first to demonstrate the transferability of LoRA adapters between base and pruned large language models (LLMs)
2. The proposed method shows strong compatibility across multiple architectures, with experiments conducted on LLaMA, Gemma, and Mistral models, demonstrating its general applicability.
3. Extensive experiments on BoolQ, SST-2, MRPC, RTE, and WinoGrande show that FIT-LoRA nearly matches or sometimes exceeds task-specific LoRA fine-tuning. The method scales well up to 13B-parameter models, maintains cross-architecture consistency.

**Weaknesses:**

1. The author mentions the concept of “Train once, fit anywhere” in Figure 2, but after reading the paper, it remains unclear to me why such a scenario is necessary or what specific problem it aims to address.
2. The paper assumes a workflow where fine-tuning is performed before pruning. It would be helpful if the authors could elaborate on why this order is preferred over the more conventional approach of pruning first and then fine-tuning. Moreover, the experments that compressed models fine-tuned with LoRA consistently achieve higher performance (scores) deserves further analysis or justification.
3. The paper’s approach to depth pruning, structural pruning, and sparsification relies on straightforward, intuitive operations. While this simplicity is an advantage in practical implementation, the work would be strengthened by a deeper analysis or ablation study clarifying why such simple operations are sufficient for effective knowledge transfer.

**Questions:**

Please refer to the Weaknesses section.

---

> ### Author Response · Authors · 2025-11-27
> **Responses to Reviewer 4 (ySHe)**
>
> We thank the reviewer for the thoughtful review. We address each concern point by point below.
>
> > “The author mentions the concept of “Train once, fit anywhere” in Figure 2, but after reading the paper, it remains unclear to me why such a scenario is necessary or what specific problem it aims to address.”
>
> **Response:** We thank the reviewer for asking the clarifying question about the concept “Train once, fit anywhere” we described in Figure 2. Such a scenario is necessary to avoid the burden of fine-tuning again on compressed base models. We revised the manuscript to add the following text in the introduction:
>
> “Our approach can help practitioners who may have a base model and many compressed models that require personalization, which would previously necessitate fine-tuning on every single model version. We provide a detailed discussion on several real-world use cases in Appendix C.”
>
> > “The paper assumes a workflow where fine-tuning is performed before pruning. It would be helpful if the authors could elaborate on why this order is preferred over the more conventional approach of pruning first and then fine-tuning.”
>
> **Response:** We thank the reviewer for asking for clarification on the ordering of the workflow.
> There are two main reasons why fine-tuning may be performed before pruning.
>
> 1. Fine-tuning has already been conducted by a model company, organization, or individual who distributes the LoRA adapter on a cloud platform such as Hugging Face for the particular base LLM. Users would like to use the LoRA adapter, but not necessarily use the base LLM as is. For example, the base LLM may be prohibitively large for the user’s intended use case, so the user leverages pruning to modify the base model. Fit-LoRA can be used to seamlessly transfer the fine-tuned knowledge from the base LLM for effectively no cost. Please also see our response to Reviewer fccj for a breakdown of motivations for data-free and training-free transfer.
> 2. While pruning first and then fine-tuning is a valid workflow, it has a significant computational disadvantage, especially when many derived models are needed and many fine-tuning customization adapters are required. The total amount of fine-tuning adapters required would be = # of pruned models $\times$ # of fine-tuning tasks. Instead, we should be able to maintain a single LoRA adapter for each task and then apply it to any derived model. This avoids expensive repeated fine-tuning and reduces disk storage cost since we only obtain one adapter per task.
>
> We revised the manuscript to add the following text in the Introduction:
>
> “Additionally, downstream users of the LoRA adapter (e.g., adapters from platforms like HuggingFace) may want to use the adapter on a compressed version of the base model. However, they do not have access to the training data or may not have resources for fine-tuning.”

---

> ### Author Response · Authors · 2025-11-27
> **Responses to Reviewer 4 (ySHe) Part 2**
>
> > “Moreover, the experiments that compressed models fine-tuned with LoRA consistently achieve higher performance (scores) deserves further analysis or justification.”
>
> > “The paper’s approach to depth pruning, structural pruning, and sparsification relies on straightforward, intuitive operations. While this simplicity is an advantage in practical implementation, the work would be strengthened by a deeper analysis or ablation study clarifying why such simple operations are sufficient for effective knowledge transfer.”
>
> **Response:** We appreciate the reviewers’ suggestion to provide justification for Fit-LoRA. We share the reviewer’s point of view that the simplicity of our approach is seen as an advantage and provides value in its ease of adoption. We have overhauled the explanation for why Fit-LoRA works at the end of Section 3, quoted as follows:
>
> “The design of our approach exploits the fact that the pruned model's layers are aligned with those of the base model. This is true in both examples of LLM-Streamline and ReplaceME, where the pruning algorithm keeps the surviving layer weights unchanged while incorporating a healing block. The healing block is trained to reconstruct the hidden outputs by the group of eliminated layers, ensuring that the surviving layers continue to receive the same input regardless from the pruned layers or the healing block. Since the weights of the surviving layers in the compressed models are unchanged, applying LoRA updates to the surviving layers can preserve most of the task adaptation signal. The only mismatch concerns the LoRA updates from the eliminated layers, which are skipped for the healing block. Empirically, we find that this copy-and-trim strategy is remarkably effective, revealing that task-specific LoRA knowledge is robust to architectural reduction.
>
> We note that the structural correspondence between the pruned and base model is crucial for LoRA adapter transferability. Our approach in transferring adapters may not be suitable for compression algorithms that do not follow this assumption. For instance, pruning methods that disrupt layerwise independence would generally not be conducive for LoRA transferability.”

---

### Official Review · Reviewer_SPkr · 2025-10-28

**Soundness:** 1
**Presentation:** 2
**Contribution:** 1
**Rating:** 0
**Confidence:** 5

**Summary:**

The paper proposes a method which combines in a straightforward manner previous methods performing Low-Rank Adaptation and different types of pruning (structured, unstructured, layer). Comprehensive experiments are conducted, which however do not show a clearly consistent benefit of the combined approach.

**Strengths:**

- Comprehensive experimental evaluation on many tasks.

**Weaknesses:**

- Straightforward combination of existing approaches.
 - No clear benefit experimentally.

**Questions:**

- What is the distinctive contribution of the method, except a straightforward combination of prior ones?

---

> ### Author Response · Authors · 2025-11-27
> **Responses to Reviewer 3 (SPkr)**
>
> > “Straightforward combination of existing approaches.”
> > “What is the distinctive contribution of the method, except a straightforward combination of prior ones?”
>
> **Response:** We thank the reviewer for the comment. Can the reviewer specify which existing approaches they are referring to?
>
> Our distinctive contribution is the insight that LoRA adapters can be transferred across pruning scales, addressing an unstudied challenge in the literature. We provide evidence for this through the simple Fit-LoRA mechanism and with extensive experiments on different pruning methods and LLM architectures.
>
> > “No clear benefit experimentally. [Edited Nov. 12]: It can be seen that LoRA FT in general provides better performance compared to the propose FIT LoRA.”
>
> **Response:** We agree that LoRA FT in general provides similar or better performance compared to Fit-LoRA. However, the reviewer has missed the main point of the work.
>
> The clear experimental benefit of Fit-LoRA is described in the originally submitted manuscript:
>
> “...enables fine-tuning knowledge transfer between a base LLM and derived LLMs of smaller scales without needing any training or access to the original fine-tuning data.”
>
> “...training-free and data-free characteristics, offers a valuable and practical way to achieve comparable task-specific performance on a variety of compressed models compared with direct task-specific fine-tuning.”
>
> “Beyond the computational and logistical burden, updates made to a base model also present another challenge: the potential lack of access to the original dataset that was used for fine-tuning.”

---

### Official Review · Reviewer_z1W4 · 2025-11-02

**Soundness:** 4
**Presentation:** 3
**Contribution:** 2
**Rating:** 6
**Confidence:** 5

**Summary:**

Authors propose Fit-LoRA, which demonstrate that LoRA adapters can transfer between backbones before and after pruning.

**Strengths:**

1. Paper is clearly written and generally easy to follow.
2. The method is simple and intuitive, and this paper conduct experiments to validate this method. Although the method itself may not be new/novel, this work constitutes a good-to-know report that community would find beneficial.
3. Experiments are solid.

**Weaknesses:**

1. Authors did not consider quantization. If considering quantization as “pruning bits”, it indeed makes sense to add quantization experiment in. For example, applying LoRA trained on Qwen3-4B (BF16) to Qwen3-4B-FP8.
2. Regarding framing, I don’t consider this work as “proposing a novel approach”, although it’s framed this way. I find the work’s value in exploring LoRA’s transferability across scales/pruning. I suggest authors frame their work as more of a valuable empirical observation like [1].
[1] LoRA Without Regret. Schulman et al. 2025

**Questions:**

[1] explores cross-scale weight parameter transfer for vision transformer pretraining. They proposed weight selection and also discuss the case where dimension and depth mismatch. I encourage authors to discuss the connection of their paper with this paper. For example, could authors transfer LoRA from LLaMA-3.1-8B to LLaMA-3.2-1B?

[2] discusses the transferability of finetuning task vector between different checkpoints of an LLM. I find the reason behind the working mechanism of Fit-LoRA is also related.

[1] Initializing Models with Larger Ones. Xu et al. ICLR 2024
[2] Param  for Direct Weight Mixing: Post-Train Large Language Model at Zero Cost. Cao et al. ICLR 2025

---

> ### Author Response · Authors · 2025-11-27
> **Responses to Reviewer 2 (z1W4)**
>
> We thank the reviewer for the thoughtful review and are encouraged that you recognized the strong empirical validation of our work. We address your concerns point by point below.
>
> > “Authors did not consider quantization. If considering quantization as “pruning bits”, it indeed makes sense to add quantization experiment in. For example, applying LoRA trained on Qwen3-4B (BF16) to Qwen3-4B-FP8.”
>
> **Response:** We thank the reviewer for the suggestion to include quantization as one of the compression scenarios. In deciding the scope of our work, we choose to focus on techniques under the family of pruning (e.g., depth, structured, and unstructured) as they introduce structural changes to the base model. In the Related Works section of the originally submitted manuscript, we wrote "... we focus on pruning. Unlike quantization, pruning often introduces architectural changes to the pretrained LLM, which makes LoRA adapter transferability challenging." We agree with the reviewer that quantization would be an interesting addition if the focus is not limited to structural changes.
>
> Additionally, it may be of interest to know that in Appendix B of the original manuscript, we have results on training a QLoRA adapter, where the adapter was trained on an nf4-bit quantized model while the LoRA adapter weights were kept in full precision of fp16 (Dettmers et al., 2023). We took the LoRA adapter and used our simple mapping method Fit-LoRA to transfer the adapter to different pruned models. The results show that QLoRA adapters are also robust enough to handle transferability.
>
> [4] QLoRA: Efficient Finetuning of Quantized LLMs. Dettmers et al. NeurIPS 2023
>
>
> > “Regarding framing, I don’t consider this work as “proposing a novel approach”, although it’s framed this way. I find the work’s value in exploring LoRA’s transferability across scales/pruning. I suggest authors frame their work as more of a valuable empirical observation like [1]. [1] LoRA Without Regret. Schulman et al. 2025”
>
> **Response:** We thank the reviewer for finding value in our work and for the re-framing suggestion. We plan to edit the paper to present our work as an empirical study in exploring LoRA’s transferability across scales.
>
>
> > “[1] explores cross-scale weight parameter transfer for vision transformer pretraining. They proposed weight selection and also discuss the case where dimension and depth mismatch. I encourage authors to discuss the connection of their paper with this paper. For example, could authors transfer LoRA from LLaMA-3.1-8B to LLaMA-3.2-1B?”
>
> **Response:** The reviewer makes an excellent point that using weight parameter transfer for initialization is a nice advantage. We distinguish our work in that we are focused on weight transfer for personalization without requiring any further training, instead of weight transfer for initialization. We added a new subsection under Related Works of the revised manuscript to discuss the connection between our work and Xu et al. (2024). We quote the added text below:
>
> “Xu et al. (2024) introduce a recipe for initializing smaller ViT models from a large ViT model via weight transfer. It consists of first-N mapping for layers, one-to-one mapping for components, and uniform mapping for elements. While our approach shares the spirit of weight selection as in Xu et al. (2024), instead of pretraining model initialization,
> our goal is to enable the immediate use of the smaller model without additional training.”
>
> We can hypothesize that Fit-LoRA can also provide a good initialization point for LoRA adapters, however, we aim to showcase capable training-free zero-shot transfer performance.
>
> [1] Initializing Models with Larger Ones. Xu et al. ICLR 2024

---

> ### Author Response · Authors · 2025-11-27
> **Responses to Reviewer 2 (z1W4) Part 2**
>
> > “[2] discusses the transferability of finetuning task vector between different checkpoints of an LLM. I find the reason behind the working mechanism of Fit-LoRA is also related.”
>
> **Response:** We agree with the reviewer that the transferability of using a task vector on another checkpoint of an LLM (Cao et al., 2025) shares a connection with our work on transferability across scales. In Cao et al. (2025), the authors use a task vector to capture the knowledge of a post-training run, and similarly in PortLLM (Khan et al., 2025), the authors use a LoRA adapter patch (can also be considered a task vector) to capture the knowledge of a fine-tuning run. We agree that the working mechanism behind these methods is similar to ours where the downstream fine-tuning data is “compressed” into a portable, task vector, or adapter, and then can be applied to a derived base model. While, Cao et al. (2025) and Khan et al. (2025) show an emergent property that adapters are resilient to temporal drift, our work shows resilience across scales. In the revised manuscript, we also add a discussion of the connection between our work and Cao et al. (2025) in Related Work. We quote the added text below:
>
> "Post-training weight transfer between LLMs to transfer instruction tuning, supervised-fine-tuning, and fine-tuning knowledge has been explored in Param∆ (Cao et al., 2025). As aforementioned, PortLLM (Shahroz et al., 2025) also shares the goal of fine-tuning knowledge transfer. In both works, the authors propose to use the difference in weights between the base and post-trained models and apply the task adapter to a new model checkpoint. However, both apply their methods to homologous models only, where there is no architectural change in the models.
> "
>
> [2] Param∆ for Direct Weight Mixing: Post-Train Large Language Model at Zero Cost. Cao et al. ICLR 2025
>
> [3] PortLLM: Personalizing Evolving Large Language Models with Training-Free and Portable Model Patches. Khan et al. ICLR 2025

---

### Official Review · Reviewer_fccj · 2025-11-09

**Soundness:** 2
**Presentation:** 3
**Contribution:** 2
**Rating:** 2
**Confidence:** 3

**Summary:**

This paper propose a training-free framework, Fit-LoRA, that aims to modify adapters (specifically LoRA) when we compress the base models with a finetuned LoRA. The paper mainly considers depth pruning, structured pruning, and sparsification for the base model compression part, and Fit-LoRA is effectively a direct selection/match over depth, channel, and no change for sparsification. The experiment is conducted on

**Strengths:**

- The paper is easy to follow and the overall idea of Fit-LoRA is understandable upon first read.
- Fit-LoRA are tested with multiple modern LLMs and compression algorithms such as ReplaceME and MaskLLM.

**Weaknesses:**

- Although training-free, the approach of Fit-LoRA is too simple if not trivial. In this case, I won't consider Fit-LoRA have a major methodology innovation as the motivation / explanation behind Fit-LoRA on line 236-240 is also insufficient and ungrounded. For example, we could have a structured compression algorithm that is not layerwise independent (for example, each layer corrects the error of a previous pruned layer), and in this case, each layer output is not necessarily aligned with prior base model. The compression algorithm can still perform well, but not necessarily Fit-LoRA. **The effectiveness of Fit-LoRA is implicitly dependent on the compression algorithm**.

- **The requirement of being training-free is much more restrictive than the requirement of cannot finetune with proprietary data, and such requirement is unrealistic.** We often still have online update or cheap calibration with auxiliary public data and if the performance degradation from compression is concerning, a cheap calibration to recover the performance loss is better than do-nothing.

- The benchmarks SST2, BoolQ, RTE, MRPC, WinoGrande are all too *easy* for modern LLM. For example, Llama3 can have zeroshot acc of SST2 as 90% and a finetuning acc of 95% is of marginal improvement. **The commonsense (Arc-C, PIQA, OBQA, etc.), arithmetic and math reasoning (GSM8K or similar) should be definitely added.**
  - This might have *major* influence on the benchmark performance of Fit-LoRA. **It is quite likely that the simplicitiy of Fit-LoRA can work with easy benchmarks but will suffer from serious degradation for harder ones.**

These 3 weaknesses are critical for the validity and generalizability of Fit-LoRA so I will vote for reject.

**Questions:**

- I am not confident that there are absolutely *no* prior works on modifying adapter after base model compression. Even this holds, there are still prior works on *transferring* adapters from 1 task to other task with training. A discussion of such should be added to the related works, or if appropriate, as a baseline to the experiment.

**Details Of Ethics Concerns:**

Not available

---

> ### Author Response · Authors · 2025-11-13
> **Part of the review comment is missing**
>
> > This paper propose a training-free framework, Fit-LoRA, that aims to modify adapters (specifically LoRA) when we compress the base models with a finetuned LoRA. The paper mainly considers depth pruning, structured pruning, and sparsification for the base model compression part, and Fit-LoRA is effectively a direct selection/match over depth, channel, and no change for sparsification. The experiment is conducted on
>
> It looks like the last part of the paragraph is missing. Is it possible for the reviewer to provide the remaining part?
>
> Thank you!

---

> ### Comment · Area_Chair_sty6 · 2025-11-13
> **Please complete the missing paragraph at your earliest convenience**
>
> Dear reviewer fccj,
>
> It would be greatly appreciated if you could complete the remaining parts of the review. Thank you for your time.
>
> --AC

---

> > ### Comment · Reviewer_fccj · 2025-11-13
> > **Update the review**
> >
> > I am sorry for not finishing the last sentence on the summary and I have added back.

---

> ### Author Response · Authors · 2025-11-27
> **Responses to Reviewer 1 (fccj)**
>
> We thank the reviewer for the thoughtful review. We address each concern point by point below.
>
> > “Although training-free, the approach of Fit-LoRA is too simple if not trivial. In this case, I won't consider Fit-LoRA have a major methodology innovation as the motivation / explanation behind Fit-LoRA on line 236-240 is also insufficient and ungrounded. For example, we could have a structured compression algorithm that is not layerwise independent (for example, each layer corrects the error of a previous pruned layer), and in this case, each layer output is not necessarily aligned with prior base model. The compression algorithm can still perform well, but not necessarily Fit-LoRA. The effectiveness of Fit-LoRA is implicitly dependent on the compression algorithm.”
>
> **Response:** We agree with the reviewer that Fit-LoRA may not be effective in scenarios where a compression algorithm is not layerwise independent. In the revised manuscript, we have explicitly stated this assumption and improved our explanation behind Fit-LoRA. We quote the edited paragraphs below:
>
> “The design of our approach exploits the fact that the pruned model's layers are aligned with those of the base model. This is true in both examples of LLM-Streamline and ReplaceME, where the pruning algorithm keeps the surviving layer weights unchanged while incorporating a healing block. The healing block is trained to reconstruct the hidden outputs by the group of eliminated layers, ensuring that the surviving layers continue to receive the same input regardless from the pruned layers or the healing block. Since the weights of the surviving layers in the compressed models are unchanged, applying LoRA updates to the surviving layers can preserve most of the task adaptation signal. The only mismatch concerns the LoRA updates from the eliminated layers, which are skipped for the healing block. Empirically, we find that this copy-and-trim strategy is remarkably effective, revealing that task-specific LoRA knowledge is robust to architectural reduction.
>
> We note that the structural correspondence between the pruned and base model is crucial for LoRA adapter transferability. Our approach in transferring adapters may not be suitable for compression algorithms that do not follow this assumption. For instance, pruning methods that disrupt layerwise independence would generally not be conducive for LoRA transferability.
> ”
>
> Although our approach of Fit-LoRA is simple, our contribution in studying cross-scale resilience of LoRA adapters and conducting large scale benchmarking on a wide range of tasks, LLMs, and multiple supported pruning methods is nontrivial.
>
>
> > “The requirement of being training-free is much more restrictive than the requirement of cannot finetune with proprietary data, and such requirement is unrealistic. We often still have online update or cheap calibration with auxiliary public data and if the performance degradation from compression is concerning, a cheap calibration to recover the performance loss is better than do-nothing.”
>
> **Response:** Training-free could be motivated by many real-world scenarios:
>
> * [No data]: Model fine-tuning requires task-specific training data that may not be available at all. Consider the following scenarios:
>    * A downstream developer wants to leverage a LoRA adapter that was distributed by a model provider or another platform. The original data used to train the adapter is not available. And of course, it would be unrealistic for a downstream developer to collect and annotate their own dataset that matches the task specification.
>    * Under privacy policies, it may be required that data collected be deleted after a certain amount of time. Data used for a previous fine-tuning run may not be available forever. There is also no “auxiliary public data” given that this type of data requires authorized access.
>    * The original data used for fine-tuning was never stored. In an online fine-tuning scenario, especially under restrictive privacy constraints where the provider cannot store user-sensitive data, we never store the data and only see it once.
>
> * [Can’t fine-tune]: Downstream developers may not have access to the GPU resources required for training. While an edge device may have the required memory for inference, it may not have enough memory for training, which is more expensive. For training, we have to store optimizer states and gradients in memory.
> * [Large cost, time and logistical savings]: More challenging to keep up with model customization as the number of deployment scenarios grows. Consider that we need to support 100s of compressed models that use different pruning techniques or have unique pruning ratios. We can also have many LoRA adapters, each for a different task. Fine-tuning each model on each task becomes cumbersome, time-consuming, and costly.
>
> We have added the detailed scenarios mentioned above in Appendix C of the revised manuscript.

---

> ### Author Response · Authors · 2025-11-27
> **Responses to Reviewer 1 (fccj) Part 2**
>
> > “The benchmarks SST2, BoolQ, RTE, MRPC, WinoGrande are all too easy for modern LLM. For example, Llama3 can have zeroshot acc of SST2 as 90% and a finetuning acc of 95% is of marginal improvement. The commonsense (Arc-C, PIQA, OBQA, etc.), arithmetic and math reasoning (GSM8K or similar) should be definitely added.
> This might have major influence on the benchmark performance of Fit-LoRA. It is quite likely that the simplicitiy of Fit-LoRA can work with easy benchmarks but will suffer from serious degradation for harder ones.”
>
> **Response:** We appreciate the reviewer’s recommendation to evaluate on arithmetic and additional commonsense benchmarks. We are actively running evaluations on these and plan to  report here soon.
>
>
> > “I am not confident that there are absolutely no prior works on modifying adapter after base model compression. Even this holds, there are still prior works on transferring adapters from 1 task to other task with training. A discussion of such should be added to the related works, or if appropriate, as a baseline to the experiment.”
>
> **Response:** We appreciate the reviewer’s comment on the uniqueness of our work in modifying the adapter after base model compression. We agree with the reviewer that there are prior works that aim to reuse adapters from one task to aid performance in **another task**. A common technique used in this line of work is LoRA merging. We added a discussion of these in the Related Works section of the revised manuscript. While prior works, such as Zhao et al. (2025) and Huang et al. (2025), aim for cross-task or multi-task generalization to boost performance on seen or unseen tasks and assume no change to the base model, our work focuses on **cross-scale** generalization for a single task.
>
> [1] Merging LoRAs like Playing LEGO: Pushing the Modularity of LoRA to Extremes Through Rank-Wise Clustering. Zhao et al. ICLR 2025
>
> [2] LoraHub: Efficient Cross-Task Generalization via Dynamic LoRA Composition. Huang et al. COLM 2024

---

### Meta-Review · Area_Chair_BZnm · 2026-01-06

**Summary:**

The paper proposes Fit-LoRA to modify LoRA adapter with different pruning methods. The reviewers all agree that the paper is well-written and the experiments are comprehensive. However, reviewers raise concerns about the novelty where the paper shows a simple and effective method but fails to give more deeper analysis. There are also some concerns about not testing more difficult reasoning benchmarks and the setting itself.

**Reviewer Concerns:**

The rebuttal partially addressed the concerns with the setting clarification and added more discussion on the related works. However, I believe the concern about clarifying the main focus is not addressed, and there are no new results on difficult reasoning benchmarks.

**Reviewer Scores:**

I believe only Reviewer ySHe may increase the review, but probably not.  Since Reviewer fccj's concerns remains unaddressed, his/her score won't be changed.

---

### Decision · Program_Chairs · 2026-01-26

Reject